# Radial Growth Responses of Four Southeastern USA Pine Species to Summertime Precipitation Event Types and Intense Rainfall Events

Tyler J. Mitchell * and Paul A. Knapp

Carolina Tree-Ring Science Laboratory, Department of Geography, Environment, and Sustainability, University of North Carolina at Greensboro, Greensboro, NC 27412, USA
* Correspondence: tjmitche@uncg.edu

**Abstract:** Previous dendroclimatic studies have examined the relationship between total precipitation amounts and tree radial growth in the southeastern USA, yet recent studies indicate that specific precipitation event types and rainfall intensities influence longleaf pine (*Pinus palustris* Mill.) radial growth unequally. It remains unknown if other pine species respond similarly regarding specific precipitation types and intensities as most dendroclimatic studies have focused on precipitation amounts on monthly-to-annual scales without examining either the event type or intensity nor focusing on daily data. Here, we examine summertime climate–radial growth relationships in the southeastern USA for four native pine species (longleaf, shortleaf, Virginia, pitch) during 1940–2020. We examine and compare each species' response to precipitation event types and intense rainfall events (IREs) and address if the temporal sensitivity to these events is species specific. Distinct temporal sensitivities exist among species, and there is a consistent association between convective, stationary front, and quasi-stationary precipitation and radial growth. All species except Virginia pine have significant ($p < 0.001$) associations between IREs and radial growth, even though IREs account for ~49% of summertime rainfall. These results suggest precipitation-type sensitivity to radial growth may have dendroclimatic implications.

**Keywords:** precipitation; pine; southeastern USA; dendroclimatology; climate; North Carolina

## 1. Introduction

The formal study of atmosphere–plant interactions is long established (e.g., [1]) and many dendroclimatic studies have examined the relationship between climatic variables and radial growth in southeastern USA pine (*Pinus* spp.) trees, including: longleaf pine (*P. palustris* Mill.) (e.g., [2–11]); shortleaf pine (*P. echinata* Mill.) [12–19]; Virginia pine (*P. virginiana* Mill.) [20]; and pitch pine (*P. rigida* Mill.) [20–23]. Generally, abundant current- and prior-year precipitation and low-temperature stress are associated with increases in radial growth in these species. However, less is known about the possible influence on radial growth based on how precipitation is received (i.e., duration, amount, and intensity) by trees. Recent work indicates that specific precipitation event types influence longleaf pine radial growth unequally. For example, longleaf pine growing in the coastal plain physiographic region of Florida and North Carolina principally respond to tropical cyclone precipitation (e.g., [8,24–26]), while longleaf pine growing in the piedmont physiographic region principally respond to a combination of convective and stationary front precipitation (hereafter, "quasi-stationary precipitation", [11]). Thus, regardless of precipitation event type, the influence of infrequent high-intensity rainfall events appears to be the leading modulator of longleaf pine radial growth, rather than total precipitation [27], even when intense events represent less than half of the total rainfall amounts, which raises the question if similar conditions exist for other pine species of the southeastern USA.

Radial growth can be broadly defined in three different terms that capture growth during distinct periods. Earlywood is radial growth produced during the early growing season (i.e., spring and early summer, [28]), latewood is produced later in the growing season (i.e., late summer to early autumn, [28]), and totalwood is the combination of earlywood and latewood produced in one annual period of growth. Finer-scaled measures of radial growth can provide greater detail in climate–growth analyses as the distinct month(s) of precipitation that best influence radial growth can be determined. For example, in the Pacific Northwest, USA annual variability of totalwood growth of western juniper (*Juniperus occidentalis* Hook.) closely reflects (+) cool-season precipitation variability [29] while for western Montana, USA totalwood growth for ponderosa pine (*Pinus ponderosa* Laws.) was most strongly associated (+) with late spring–midsummer precipitation [30]. In eastern Colorado, USA ponderosa pine adjusted latewood growth best corresponds to the highest 1-day precipitation amount in a 2-week period in July (+) [31]. Further, a combination of tree species sourced from the International Tree-Ring Data Bank best respond to distinct seasonal and sub-seasonal precipitation amounts in North America [32] and the USA west coast [33], with improved model skill and specificity when compared to either total precipitation or totalwood measurements. In the southeastern USA, a focus on latewood radial growth would isolate the environmental influences that are active during the late summer to early autumn as latewood growth of these four southeastern pine species express substantial interannual variability (ranging from 40% (longleaf pine) to 44% (shortleaf pine, Virginia pine) of totalwood) and has a strong association with summer moisture conditions (e.g., [5,18,19]).

The four pine species are both temperature and moisture sensitive throughout their ranges, and studies have examined the influence of climate on latewood radial growth. Longleaf pine latewood is associated with previous August–September maximum temperature (+), current July–October maximum temperature (−), current March–October precipitation (+), previous August–December precipitation (−), current August–September PHDI (+), and previous September–December PHDI (−) in south Alabama, USA [5], June–October PDSI (+) and precipitation (+) along a physiographic gradient in North Carolina [9], and summer–fall precipitation in Texas and South Carolina [7]. Shortleaf pine latewood is associated with July–September precipitation (−) and temperature (+) in Arkansas [13], July–September PDSI (+) and July precipitation (+) in central North Carolina [19], and June–September precipitation (+) and temperature (−) in eastern Oklahoma [18]. Pitch pine latewood growth is associated with previous July and current August precipitation (+) near the species' northern range limit in central Maine [23]. Comparatively less research has examined climate–growth relationships of Virginia pine (e.g., [20]), likely because the lack of trees > 100 years old limits the species' value for dendroclimatic studies. To our knowledge, no studies have examined relationships between Virginia pine latewood and climate.

Results from finer-scale analyses of precipitation–growth relationships have implications for tree ecophysiology, as the specific mode of precipitation that influences radial growth can be better understood (e.g., [34–37]) and for climate reconstructions and precipitation models using tree-ring data as an input source. Particularly, finer-scale precipitation measures may reduce the likelihood of false drought identification (i.e., a year with reduced radial growth, yet above-mean overall precipitation) inferred from tree-ring precipitation reconstructions. Understanding how different species respond to specific modes of precipitation can increase model specificity and provide information about interspecific responses to drought, or a lack of intense rainfall events, which has been demonstrated as an area in need of further research (e.g., [38]). It remains unknown if other pine species respond similarly to longleaf pine regarding specific precipitation types and intensities as most dendroclimatic studies in the southeastern USA have focused on precipitation amounts on monthly-to-annual scales without examining either the event type or intensity nor focusing on daily data. Thus, to better understand climate–radial growth relationships in the southeastern USA, our objectives were to examine and compare how each species

responds to: (1) distinct precipitation event types and (2) intense rainfall events. Additionally, we address if (3) the temporal sensitivity to these events varies among species during the summer.

## 2. Materials and Methods

### 2.1. Study Area

2.1.1. Uwharrie Mountains/Uwharrie National Forest

We sampled longleaf, shortleaf, and Virginia pine in Uwharrie National Forest in central North Carolina, USA (35.411, −80.061) growing on steep south-facing slopes to maximize climate sensitivity. The Uwharrie National Forest elevation ranges between 150 and 250 m with 15–45% slopes and "extreme bouldery" soil (NRCS Soil Survey Staff 2019, Figure 1), factors which co-vary to produce a strong climate–radial growth relationship on the landscape [10]. The most prevalent coniferous species in the Uwharrie Mountains are shortleaf pine and Virginia pine, with remnant (i.e., ~60 ha, [39]) longleaf pine populations dispersed throughout the forest on exposed south-facing slopes. The sampled trees are located in an area of historic fire suppression with no prescribed burning until 2010 [11].

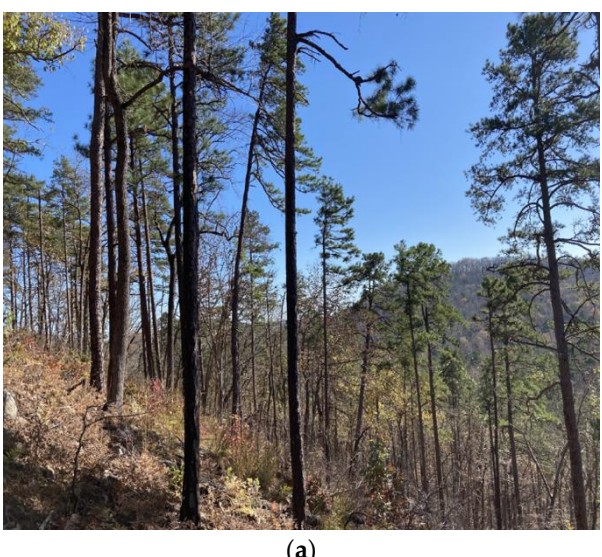
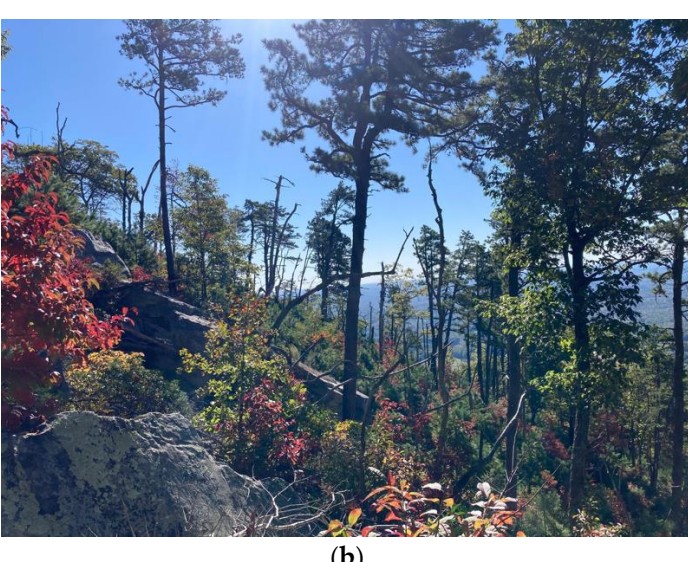

(**a**)               (**b**)

**Figure 1.** Rocky south-facing slope in (**a**) the Uwharrie Mountains, Uwharrie National Forest, NC, USA; and (**b**) the Sauratown Mountains, Pilot Mountain State Park, NC, USA. Longleaf, shortleaf, and Virginia pine co-occur in (**a**), and pitch pine occurs in (**b**).

2.1.2. Sauratown Mountains/Pilot Mountain State Park

We sampled pitch pine on south-facing slopes in Pilot Mountain State Park in central North Carolina, USA (36.339, −80.474). The Sauratown Mountains range between 300 and 780 m in elevation with 25–90% slopes characterized by "stony", and "rubbly", to "extreme bouldery" soils (NRCS Soil Survey Staff 2021, Figure 1). Pitch pine is the most prevalent pine species on steep slopes except in areas of rocky outcrops where lower fuel densities provide a more suitable habitat for Table Mountain pine. The sampled trees are located in an area of historic fire suppression (i.e., ~1930–2014, [40]), which reduced fire recurrence frequency by approximately 50% from ~2.4 years to ~4.5 years [40].

### 2.2. Tree-Ring Data

At each location, we collected two core samples per tree using a 5.15 mm diameter increment borer following standard dendrochronological sampling techniques [28]. Additionally, we recorded stem diameter, crown height, and geographic location for each tree. After field data collection, we processed and sanded samples with progressively finer grit sandpaper until the cellular structure was apparent. We scanned samples at high

resolution (i.e., DPI ≥ 1200), measured at 0.001-mm precision using WinDENDRO (Regent Instruments Inc., Quebec, Canada 2013), and crossdated. We assessed crossdating accuracy using COFECHA [41] and we created standardized chronologies using ARSTAN [41] and dplR [42] using negative exponential detrending.

We created latewood chronologies and based our analyses on latewood as it is the most responsive to climate in the study area when compared to either earlywood or totalwood [10,43]. We used adjusted latewood chronologies for analyses to account for the influence of earlywood on latewood variance ("adjusted latewood" [44]). We created full and reduced chronologies for each species based on each standardized sample's response to precipitation. The reduced chronology is comprised of the most climatically sensitive trees (i.e., a significant, $p < 0.05$, relationship to precipitation).

### 2.3. Climate Data

We selected daily precipitation data during 1940–2020 from weather stations with high daily data completeness (i.e., >95%) that are located nearby to sampling locations (Figure 2). We selected Albemarle, North Carolina (Station ID: USC00310090), Randleman, North Carolina (USC00317097), and Siler City, North Carolina (USC00317924) as the Uwharrie Mountain stations and Mount Airy, North Carolina (Station ID: USC00315890) and Yadkinville, North Carolina (USC00319675) as the Pilot Mountain stations (Figure 2). We visually classified precipitation event types similar to prior research [11] using daily weather maps from NOAA (available online at: library.noaa.gov/Collections/Digital-Collections/US-Daily-Weather-Maps, accessed on 10 September 2022) to classify each day with precipitation as either convective (no visible frontal or tropical activity), frontal (cold, stationary, warm), or tropical (a tropical cyclone within 223 km of the study area as indicated by IBTrACS [11,45]).

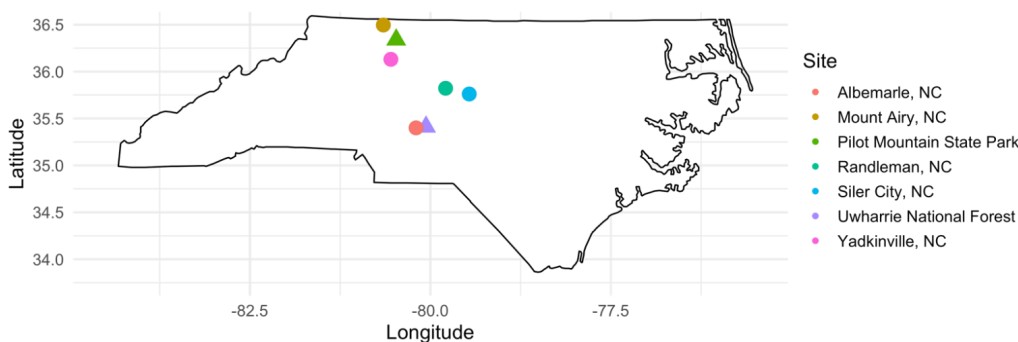

**Figure 2.** Location of the field collection sites (triangles), Uwharrie National Forest (purple triangle) and Pilot Mountain State Park (green triangle), and weather station sites (circles) in North Carolina (NC), USA.

We calculated intense rainfall event (hereafter, "IRE") frequency and precipitation amounts as days where the 24 h precipitation total was >2.0 standard deviations above the daily mean for each station for each selected month, which corresponds to approximately the 90th percentile of the distribution [27]. We separated precipitation data into: (1) total precipitation; (2) precipitation associated with IREs; and (3) non-IRE precipitation (i.e., precipitation$_{total}$ − precipitation$_{IRE}$) and calculated the mean values of these variables from the mean values from three Uwharrie Mountain stations and the two Pilot Mountain stations for use in analyses. Additionally, we created low, medium, and high-IRE frequency groupings for each species to analyze each species' radial growth responses during these years.

### 2.4. Statistical Analyses

We tested the relationship between combinations of single and multiple months of precipitation data with the tree-ring chronologies using Pearson's correlation coefficient

to select the month(s) which best corresponded to radial growth for the four species. Once the most responsive month was selected, we tested the association between specific precipitation event types and IRE frequency and precipitation and radial growth for each species for that (those) selected month(s). We tested for significant differences in radial growth for each species during years with low, medium, and high-IRE frequency using one-way ANOVA with Tukey's HSD test.

## 3. Results and Discussion

The full longleaf pine chronology is comprised of 80 samples from 60 trees with a series intercorrelation of 0.576, a mean sensitivity of 0.497 spanning 1740–2020, and the full shortleaf pine chronology is comprised of 47 cores from 31 trees with a series intercorrelation of 0.592, a mean sensitivity of 0.428 spanning 1775–2020. The Virginia pine chronology is comprised of 20 cores from 13 trees with a series intercorrelation of 0.439, a mean sensitivity of 0.359 spanning 1929–2020, and the pitch pine chronology is comprised of 49 cores from 27 trees with a series intercorrelation of 0.626, a mean sensitivity of 0.435 spanning 1844–2020. The reduced chronologies contain 64 samples from 47 trees (longleaf pine, interseries = 0.584, mean sensitivity = 0.502, 1740–2020), 34 samples from 25 trees (shortleaf pine, 0.597, 0.437, 1775–2020), and 35 samples from 22 trees (pitch pine, 0.632, 0.446, 1844–2020). The Virginia pine sample size was too small for a reduced chronology to be used for analyses.

The longleaf pine ($r = 0.65$, 95% CI [0.51, 0.76], $p < 0.001$) and shortleaf pine ($r = 0.59$, 95% CI [0.42, 0.71], $p < 0.001$) chronologies best respond to the same months of precipitation (i.e., July–September) during 1940–2020 suggesting similar temporal sensitivities between the two species. The Virginia pine chronology responds best to August–September precipitation ($r = 0.37$, 95% CI [0.17, 0.55], $p < 0.02$) while pitch pine responds to August precipitation ($r = 0.40$, 95% CI [0.20, 0.57], $p < 0.002$). Generally, more variance is explained when using the reduced chronologies for longleaf ($r = 0.59$, 95% CI [0.42, 0.71], $p < 0.001$), shortleaf ($r = 0.63$, 95% CI [0.47, 0.74], $p < 0.001$), and pitch pine ($r = 0.51$, 95% CI [0.33, 0.65), $p < 0.001$). The reduced chronologies consistently respond to either convective (longleaf pine, shortleaf pine), stationary front precipitation (all pine species), or a combination of convective and stationary front precipitation, quasi-stationary precipitation (all pine species) (Table 1), which is consistent with previous work for longleaf pine [11] and emphasizes the importance of precipitation intensity on radial growth. No species significantly responded to tropical cyclone precipitation.

All pine chronologies except Virginia pine have significant ($p < 0.001$) relationships with IRE precipitation, which on average represents 49.2% of total precipitation amounts (Figure 3, Table 2) and is most commonly associated with stationary front precipitation events. Virginia pine has a near significant ($p = 0.052$) relationship with August–September IRE precipitation ($r = 0.32$, 95% CI [0.11, 0.51], $p = 0.052$), a coefficient that, while non-significant, is similar to its overall precipitation response ($r = 0.37$). Longleaf pine ($r = 0.55$, 95% CI [0.37, 0.68], $p < 0.001$) and shortleaf pine ($r = 0.54$, 95% CI [0.36, 0.68], $p < 0.001$) adjusted latewood is associated with July–September IRE precipitation while pitch pine is associated with August IRE precipitation ($r = 0.50$, 95% CI [0.31, 0.65], $p < 0.001$). Non-IRE precipitation (i.e., precipitation$_{total}$ − precipitation$_{IRE}$) is not significantly associated with the pine chronologies during 1940–2020, emphasizing a distinct association between IRE rather than total precipitation (Figure 3, Table 2). Further, the use of IRE precipitation in climate–growth analyses produces a similar response to total precipitation measures and ranges from representing 98.0% (pitch pine, Table 2) to 85.7% (shortleaf pine, Table 2) of the total precipitation correlation coefficient, despite accounting for approximately half of the total precipitation amount. Mechanistically, we posit that IRE precipitation is more efficient at saturating the soil-moisture column at depth compared to lower-rainfall amount events that may only superficially increase soil moisture. These results particularly emphasize the influence of IRE precipitation on tree radial growth and suggest that climate–

growth models, and their resultant climate reconstructions, should account for precipitation intensity rather than total precipitation amounts.

**Table 1.** Correlations between convective, stationary front, and quasi-stationary precipitation with each species and the ratio of quasi-stationary precipitation to total precipitation during 1940–2020. Quasi-stationary precipitation is the combination of convective and stationary front precipitation.

| Species * | Correlation with Convective Precipitation | Correlation with Stationary Front Precipitation | Correlation with Quasi-Stationary Precipitation | Quasi-Stationary Precipitation/Total Precipitation Ratio |
|---|---|---|---|---|
| Longleaf pine | 0.35 [0.14, 0.53] $p < 0.05$ | 0.40 [0.19, 0.57] $p < 0.01$ | 0.50 [0.31, 0.65] $p < 0.005$ | 86.4% |
| Shortleaf pine | 0.40 [0.19, 0.57] $p < 0.01$ | 0.44 [0.24, 0.61] $p < 0.01$ | 0.54 [0.36, 0.68] $p < 0.005$ | 86.4% |
| Virginia pine | 0.17 [−0.06, 0.38] $p > 0.99$ | 0.36 [0.15, 0.54] $p < 0.03$ | 0.41 [0.20, 0.58] $p < 0.01$ | 83.0% |
| Pitch pine | 0.26 [0.03, 0.46] $p = 0.568$ | 0.37 [0.15, 0.55] $p < 0.03$ | 0.40 [0.19, 0.57] $p < 0.01$ | 90.3% |

\* All *r*-values are for July–September for longleaf and shortleaf, August–September for Virginia, and August for pitch.

**Table 2.** Correlations with total precipitation, IRE precipitation, non-IRE precipitation for each species, and the proportion of total precipitation correlation coefficient explained by IRE precipitation. Ratio of IRE precipitation to total precipitation during the selected month(s).

| Species * | Correlation with Precipitation | Correlation with IRE Precipitation | Correlation with Non-IRE Precipitation | IRE Precipitation/Total Precipitation Ratio | IRE Correlation/Total Precipitation Correlation |
|---|---|---|---|---|---|
| Longleaf pine | 0.59 [0.42, 0.71] $p < 0.001$ | 0.55 [0.37, 0.68] $p < 0.001$ | 0.19 [−0.03, 0.39] $p > 0.99$ | 50.2% | 93.2% |
| Shortleaf pine | 0.63 [0.47, 0.74] $p < 0.001$ | 0.54 [0.36, 0.68] $p < 0.001$ | 0.28 [0.06, 0.47] $p = 0.345$ | 50.2% | 85.7% |
| Virginia pine | 0.37 [0.17, 0.55] $p < 0.02$ | 0.32 [0.11, 0.51] $p = 0.052$ | 0.18 [−0.04, 0.38] $p > 0.99$ | 53.3% | 86.5% |
| Pitch pine | 0.51 [0.33, 0.65] $p < 0.001$ | 0.50 [0.31, 0.65] $p < 0.001$ | 0.24 [0.02, 0.43] $p = 0.262$ | 43.2% | 98.0% |

\* All *r*-values are for July–September for longleaf and shortleaf, August–September for Virginia, and August for pitch.

There are significant differences in radial growth for each species in summers with different IRE frequencies during 1940–2020. For each species, group thresholds were established based on the number of months of precipitation to which the species responded significantly. For longleaf pine and shortleaf pine, low- (years with <3 IREs), medium- (≥3 but ≤5), and high-frequency (>5) groupings were used to classify each summer during 1940–2020. For Virginia pine, low-(<2), medium (≥2 but ≤4), and high-frequency (>4) groupings, and for pitch pine, low (<1), medium (≥1 but ≤3), and high-frequency (>3) groupings were selected. The three pine species growing in the Uwharrie Mountains have significant differences during low- and high-IRE frequency years (Figure 4) such that radial growth is reduced by −0.298 (−0.465––0.131 95% CI, longleaf pine), −0.234 (−0.379––0.089 95% CI, shortleaf pine), and −0.190 (−0.370––0.010 95% CI, Virginia pine) during low-IRE frequency years. Additionally, longleaf pine radial growth is significantly

different during medium- and high-IRE frequency years (Figure 4a) such that growth is reduced by −0.203 (−0.346–−0.061 95% CI) during medium-IRE frequency years. Pitch pine also exhibits this difference between medium- and high-IRE frequency years (Figure 4d) such that radial growth is reduced by −0.153 (−0.282–−0.025 95% CI) during medium-IRE frequency years.

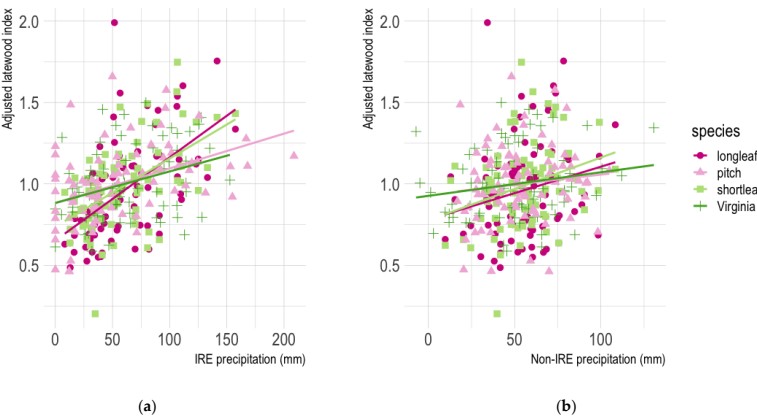

**Figure 3.** Scatterplot of adjusted latewood, IRE precipitation (**a**), and non-IRE precipitation (**b**) for each species during 1940–2020. All pine species except Virginia pine have significant ($p < 0.001$) associations with IRE precipitation (**a**) while no species have significant associations with non-IRE precipitation (**b**). No significant ($p < 0.05$) trends in either IRE or non-IRE precipitation occurred during the study period.

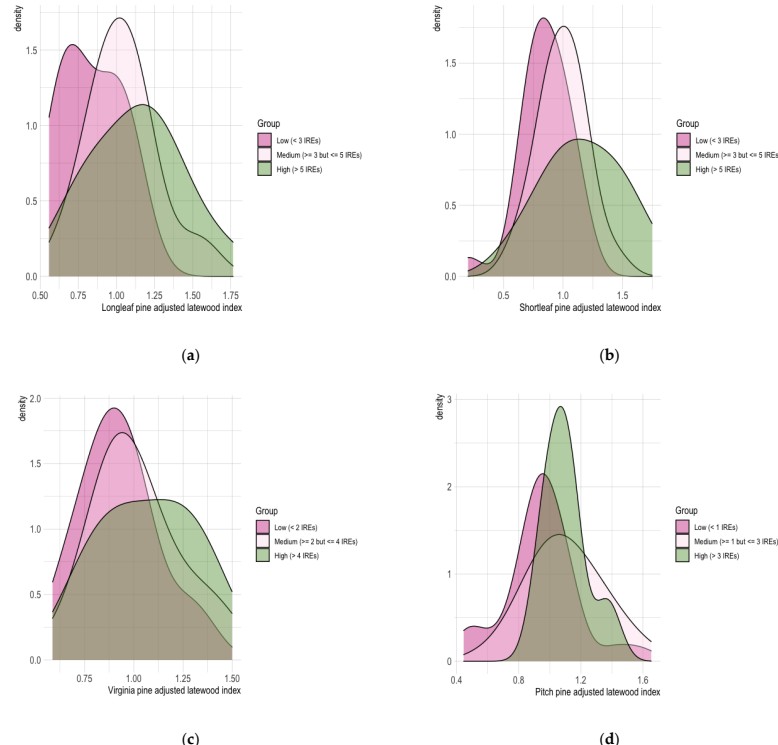

**Figure 4.** Adjusted latewood (*x*-axis) distribution density (*y*-axis) during low-, medium-, and high-IRE frequency summers for (**a**) longleaf pine, (**b**) shortleaf pine, (**c**) Virginia pine, and (**d**) pitch pine. The three pine species growing in the Uwharrie Mountains (**a**–**c**) have significant growth reductions during low-IRE frequency years when compared to high-IRE frequency years displayed by a significant distribution density shift. Longleaf pine (**a**) and pitch pine (**d**) have significant growth reductions during medium-IRE frequency years when compared to high-IRE frequency years.

## 4. Conclusions

Here, we examined how four pine species native to the southeastern USA responded to distinct precipitation event types and intense rainfall events. We found that precipitation event types are distinctly associated with radial growth with a consistent association between convective (longleaf pine, shortleaf pine), stationary front, and quasi-stationary (all pine species) precipitation, emphasizing the importance of a focus on finer-scaled measures of precipitation when analyzing climate–growth relationships. All species, except Virginia pine, have a significant ($p < 0.001$) relationship with IRE precipitation, which on average represents 49.2% of total summertime precipitation amounts. This finding suggests that precipitation–growth relationships in similar environments in the southeastern USA may better reflect specific types and/or intensities of precipitation, and the importance of ground-soaking rainfall events, rather than total precipitation variability. Thus, with an increasing emphasis on how trees respond to moisture deficit and drought (e.g., [38,46]), it is important for studies to accurately reflect the influence of specific types of precipitation on tree radial growth. Given the increasing frequency of IREs in the southeastern USA [47], these findings of precipitation-event type specificity can be applied to dendroclimatic reconstruction studies to place the current trend in a historical context.

**Author Contributions:** Conceptualization, T.J.M. and P.A.K.; methodology, T.J.M. and P.A.K.; software, T.J.M. and P.A.K.; validation, T.J.M. and P.A.K.; formal analysis, T.J.M. and P.A.K.; investigation, T.J.M. and P.A.K.; resources, T.J.M. and P.A.K.; data curation, T.J.M. and P.A.K.; writing—original draft preparation, T.J.M. and P.A.K.; writing—review and editing, T.J.M. and P.A.K.; visualization, T.J.M. and P.A.K.; supervision, T.J.M. and P.A.K.; project administration, T.J.M. and P.A.K. All authors have read and agreed to the published version of the manuscript.

**Funding:** This research received no external funding.

**Institutional Review Board Statement:** Not applicable.

**Informed Consent Statement:** Not applicable.

**Data Availability Statement:** The data presented in this study are available on request from the corresponding author.

**Acknowledgments:** We thank Marley Allen, Avery Catherwood, John Cline, Hunter Lewis, Thomas Patterson, and Jeffy Summers for their assistance in field data collection. We thank the three reviewers for their thoughtful comments that significantly improved the quality of the manuscript, and we thank the *Atmosphere* Editorial Team for their comments and proofreading.

**Conflicts of Interest:** The authors declare no conflict of interest.

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
