# Peer review of "Radial Growth Responses of Four Southeastern USA Pine Species to Summertime Precipitation Event Types and Intense Rainfall Events"

_atmosphere, doi:10.3390/atmos13101731_

Round 1

Reviewer 1 Report

The authors examine the response of four pine species in the southeastern U.S. to different precipitation event types and to intense precipitation events.  This paper seems to be publishable, but there are a few details left unsaid, particularly in the methods.  I do not think it will take the authors long to respond to these comments and I consider them minor. 

1)  My most significant comment involves the treatment of the station data.  The authors note three stations were selected as the Uwharrie Mountain station and two stations were selected as the Pilot Mountain stations, but there is a crucial lack of detail here.  How was the final time series from the station data at the two locations computed?  Were the stations at each location simply averaged together and then correlated against their respective latewood chronologies?  Were the intense rainfall event time series at each station also averaged?

2)  Lines 92-96:  There is evidence that ponderosa pine from eastern Colorado responds to the highest one-day rainfall over a 14-day window in late July (see Howard and Stahle 2020). 

3)  Line 171:  Change monthly and monthly to combinations of single and multiple months of precipitation data...

4)  Table 1:  The table caption may be a good place for the authors to remind readers that “quasi-stationary” is referring to both convective and stationary front precipitation given earlier statement in the introduction (line 41). 

References Mentioned In Review:

Howard and Stahle 2020:  Tree-ring reconstruction of single-day precipitation totals over eastern Colorado.  Monthly Weather Review, 148, 597-612. 

Author Response

Thank you for reviewing our manuscript, please see the attachment.

Reviewer 2 Report

In “Radial growth responses of four southeastern USA pine species to summertime precipitation event types and intense rainfall events,” the authors present a very interesting fine-scale analysis of precipitation-tree growth relationships in multiple pine species. The methods are solid and the results of this study should be of interest to dendrochronologists, climate scientists, and regional forest managers. I recommend minor edits before publication. Note the starred (**) comment.

Introduction:

Page 1, Ln 28 – A rare, but important citation [1]

Page 1, Ln 31 – Add the describer after each species name (P. palustris Mill., etc.).

Page 3, Ln 100 – Define “warm season.”

Materials and Methods:

Page 3, Ln 105 – “We collected radial growth measurements…” This is a bit awkward since you collected core samples in the field and measured radial growth in the lab. This is the same for section 2.1.2. Maybe change to “We sampled longleaf, shortleaf, and Virginia pine…”

Figure 1 – Great photos!

Page 4, Ln 131 – Here you describe collection and lab methods…

Results and Discussion:

Page 5, Ln 181 – Also note the number of individual trees in addition to “cores” and “samples.”

**Page 6, Ln 226 – This is the biggest takeaway from your work. Can you hypothesize a mechanism for this relationship between IRE and the growth of these pine species? If so, add here and in Conclusions.

Author Response

(The authors gave the same response as above.)

Reviewer 3 Report

The authors use four different pine species from the southeastern United States to examine the relationship between latewood growth and different precipitation types using daily station data from 1940-2020.  This is a relevant paper because there has been a recent push in dendroclimatology to utilize tree-ring data to estimate climate variability at finer timescales.   However, the paper lacks sufficient explanations on a number of methods and results presented and these will need to be addressed for this paper to eventually be published.  Furthermore, an important component missing from this paper are hypotheses for the mechanism(s) by which intense rainfall events (IREs) correspond better to latewood growth compared to non-IREs, even though the former only accounts for about half the precipitation total.  If these issues are sufficiently addressed, then publication is possible.   

Specific comments:

Lines 9 and 28:  The exact same wording used at the beginning of the abstract is used in the introduction. 

Lines 54-58: These references seem insufficient and somewhat arbitrary to describe all the work that has been previously done examining the relationship between sub-annual tree-ring measurements and climate variability. Why only use the Pacific Northwest and Western Montana as examples? 

Some possible relevant citations worth including (among many others):

Wise EK. 2020. Sub-Seasonal Tree-Ring Reconstructions for More Comprehensive Climate

Records in U.S. West Coast Watersheds. https://doi.org/10.1029/2020GL091598

Stahle DW, Coauthors 2020. Dynamics, Variability, and Change in Seasonal Precipitation

Reconstructions for North America.  https://doi.org/10.1175/JCLI-D-19-0270.1

Figure 1: Some type of site map would be beneficial here with both the locations of the tree-ring sites and instrumental stations, perhaps as an inset at the top right of one of the nice photos taken.  It would help to be able to visualize geographically where these sites are relative to each other and the stations.

Line 143: we use

Line 145-146:  Are the most climatically sensitive trees determined based on correlations between raw ring widths and the climate variable or is each series standardized? Which climate variable is being used in the correlations to determine the most sensitive trees?

Lines 155-159:  It’s still unclear even after reading the reference how event types are determined.  Is this done subjectively? Some type of objective classification scheme?

Line 163-164:  I could be misunderstanding how an IRE is determined, but is the distribution of precipitation computed for all possible days from 1940-2020?  Or are the distributions calculated based on the season that is best correlated with latewood growth (i.e. July-September)? If the former, isn’t it problematic to compute the distribution using precipitation that occurs in months and seasons not related to latewood growth of these species?

And to clarify for myself (which could be added to the text), a daily mean is calculated at each station and then if the daily precipitation total is 2.0 standard deviations above that mean it is considered an IRE?  Are zero values included in the calculation of the mean? 

Table 1 results:  What percent fraction does convective, stationary, and quasi-stationary contribute to the July-September total? If these calculations are being done for IRE vs. non-IRE then it should be done for event type as well.

Figure 2: These scatterplots are very hard to interpret since all four species are included in the same plot and the color scheme is problematic.  It would be better to have four separate panels for each species (so 8 panels in total). If not possible, then a different color scheme is needed and I would recommend using different symbols rather than only filled circles.  There is a lot of unneeded white space between the panels as well.    

This is a minor comment, but why are the y-axis labels at the top left of the panel rather in the middle, and the x-axis labels are at the far right?

Figure 3: Again, there is unnecessary white space in this figure.  You don’t need to plot the ‘group’ legend four times if they are the same across all panels.  

Between Figures 2 and 3, I would like to see as an example what a time series of IRE versus non IRE precipitation from July-September for 1940-2020 looks like (perhaps just for one station, or all four if space permits).  I suspect that IRE precipitation will have much higher variance than non-IRE and that in part explains why higher correlations are observed between the tree-ring data and IRE precipitation.  Nevertheless, I think it would be important to visualize these precipitation types.

Lines 266-286:  The conclusions are lacking any sufficient explanation as to why latewood growth in these pine species better corresponds to IRE precipitation versus non-IRE.  The two account for about the same percentage of the seasonal total, but it appears the correlations with latewood are significantly different.  Why?  Is it simply because of the higher variance in IRE?  Is IRE precipitation more efficient at penetrating the soil moisture column and thus more beneficial to the plant?  What about the differences observed in correlation between convective, stationary, and quasi-stationary?  All that is being presented are correlations without any physical explanation.  And some of these correlations are relatively modest, which would make it problematic down the road to try and calculate past estimates of IRE precipitation. How can we improve these correlations to make more robust reconstructions of extreme precipitation?

Minor comment:  What’s the relationship between IRE and event type? Of the three, does IRE correspond more to convective, stationary, or quasi-stationary?  I think that’s a link worth connecting in the narrative of the paper. 

Author Response

(The authors gave the same response as above.)

Round 2

Reviewer 3 Report

I believe the authors have adequately addressed the concerns raised upon initial review.  I have no further comments or recommendations.